



# Regional snow-avalanche detection using object-based image analysis of near-infrared aerial imagery

Karolina Korzeniowska[1, 2], Yves Bühler[3], Mauro Marty[4], Oliver Korup[2],

[1]3D Mapping, BSF Swissphoto GmbH, Schönefeld, 12529, Germany
[2]Geohazards Research Group, University of Potsdam, Potsdam, 14476, Germany
[3]WSL Institute for Snow and Avalanche Research SLF, Davos, 7260, Switzerland
[4]Swiss Federal Institute for Forest, Snow and Landscape Research WSL, Birmensdorf, 8903, Swizterland

*Correspondence to*: Karolina Korzeniowska (karolina.korzeniowska@bsf-swissphoto.com)

**Abstract.** Snow avalanches are destructive natural hazards in mountain regions that continue to claim lives, and cause infrastructural damage and traffic detours. Given that avalanches often occur in remote and poorly-accessible steep terrain, their detection and mapping is extensive and time consuming. Nonetheless, systematic avalanche detection over large areas could help to generate more complete and up-to-date inventories (cadastres) necessary for validating avalanche forecasting and hazard mapping. In this study, we have focused on automatically detecting avalanches and classifying them into release

zones, tracks, and runout zones based on 0.25-m near-infrared (NIR) ADS80-SH92 aerial imagery using an object-based image analysis (OBIA) approach. Our algorithm takes into account the brightness, the normalised difference vegetation index (NDVI), and the normalised difference water index (NDWI) and its standard deviation (SDNDWI) in order to distinguish avalanches from other land-surface elements. Using normalised parameters allows readily applying this method across large areas. We trained the method by analysing the properties of snow avalanches at three 4-km² areas near Davos,

Switzerland. We compared the results with manually-mapped avalanche polygons, and obtained a user's accuracy of >0.9 and a Cohen's kappa of 0.79 – 0.85. Testing the method for a larger area of 226.3 km², we estimated producer's and user's accuracies of 0.61 and 0.78, respectively, with a Cohen's kappa of 0.67. Detected avalanches that overlapped with reference data by >80% occurred randomly throughout the testing area, showing that our method avoids overfitting. Our method shows potential in large-scale avalanche mapping, although further investigations into other regions are desirable to verify

the stability of our selected thresholds and the transferability of the method.

## 1 Introduction

Snow avalanches are frequent and destructive mountain hazards, particularly during the winter and spring months. They are fast mass movements controlled by weather conditions, snowpack, and the topography of the terrain (Schweizer et al., 2003; Castebrunet et al., 2012). Avalanches can cause loss of lives, disrupt infrastructure, and affect buildings (Bründl et al., 2004;

McClung and Schaerer, 2006; Eckerstorfer and Malnes, 2015).



Despite numerous efforts aimed at reducing the risk posed by avalanches, most fatalities in Europe occur during sporting activities, caused by avalanches triggered by the victims themselves (Techel et al., 2015). Past research indicates that poor decision-making and forecasting are the main causes of deadly avalanche accidents (Techel et al., 2015; McClung, 2016). Techel et al. (2015) stated that most destructive events occur on days when the snow avalanche risk is very critical and the

snowpack layer is weak. In total, 4,750 people lost their lives in the European Alps between 1970 and 2015 (Techel et al., 2016); in the past two decades, avalanches in the Swiss Alps alone have killed 461 people (Fig. 1). Most fatal accidents have occurred in the cantons of Valais and Grison, which are the two largest in Switzerland and contain some of the highest areas in the Swiss Alps. Since 1946, avalanches in Switzerland have had the highest share of victims (37%) in comparison to other natural hazards, including lightning (16%), floods (12%), windstorms (10%), rockfalls (8%), and landslides (7%; Badoux et

al., 2016).

Avalanches killed a total of 36 people in Switzerland in the fatal winter of 1998/99 (Fig. 1). Between 27 January and 25 February, 17 people died in villages and on roads, and material losses surpassed 600 million Swiss Francs. This catastrophic winter spurred an initiative aimed at improving avalanche safety and reducing concomitant losses (Wilhelm et al., 1999; Bründl et al., 2004). This initiative included establishing an information system for exchanging data between the WSL

Institute for Snow and Avalanche Research SLF and local authorities; the development of hazard maps showing zones with high avalanche susceptibility (Bründl et al., 2004); and an increase in the artificial release of avalanches to decrease hazard levels. In this context, the need for documenting avalanches also increased, to allow for learning from past accidents.

This initiative showed clearly that regional-scale mapping of avalanches and identifying potential release zones is not only desirable, but also essential for producing avalanche cadastre maps to be used for quality-checking hazard mapping and

forecasting (Bühler et al., 2015). To date, experts (Bühler et al., 2009) map most avalanches manually, focussing mainly on geographic coordinates, but rarely on any detailed information about their extent or area. Moreover, avalanche inventories are biased toward damaging events or those reported from accessible terrain. Hence, avalanches remain notoriously underreported over larger regions. To more broadly collect information concerning avalanches, non-expert observers in Switzerland are now able to report sightings via an app (http://www.slf.ch/lawinenbulletin/rueckmeldung/index_EN), where

they can enter the location and date of their observation. Optical remote sensing data, both airborne and satellite, offer coverage that is more systematic, and are therefore increasingly used to track avalanches. Satellite images allow the collection of a picture for the same area with a time interval equal to one satellite orbit around the Earth. Airborne images can be acquired even more often, although in the winter season it is not preferable to continue the campaign due to financial aspects, because such images, which represent mostly only snow, are not convenient for any purposes other than the

assessment of the risk of snow avalanches.

Automatic methods for detecting snow avalanches are still in the developing stage (Eckerstorfer et al., 2016), and different kinds of data, such as optical and radar images, and classification approaches are used to verify their suitability to track avalanche events. This motivated us to verify the usability of near infrared (NIR) aerial images and their calculated derivatives, in mapping avalanches over a large area in Switzerland, as well as verifying the topographic conditions of their



occurrence. We have proposed an automatic OBIA-based method for detecting avalanche runout zones, as well as their tracks and release areas. We tested whether normalised indices of water and vegetation derived from aerial ADS80-SH92 images are suitable in this regard, and introduced a simple method for roughly distinguishing these zones, because knowledge concerning potential release zones critically aids in hazard assessments and runout models. Our motivation was

to develop an algorithm widely applicable to mountain regions, as relying on image spectral properties alone (Lato et al., 2012) may limit such portability, because objects may have a similar brightness to that of snow.

Most avalanches start on slopes with a median inclination of 39° (Schweizer and Jamieson, 2001), hence on slopes most difficult for skiing. We distinguish two types of avalanche: loose avalanches start from a point and gradually increase in size as they move downslope, whereas slab avalanches involve the detachment of large planar packs of snow (Fig. 2; Schweizer

et al., 2003; Bagli and Schweizer, 2009). Path length defines whether an avalanche is small (10 – 100 m), medium (100 – 1,000 m), or large (>1,000 m; Eckerstorfer et al., 2016). Most large avalanches, such as that shown in Figure 2, are slab avalanches because to bring down a large amount of snow, planar snow detachment is necessary; however, new smaller loose snow avalanches may occur and overlap with the previous one, thus complicating their detection in the field. In terms of avalanche deposit area, we distinguish between large deposits (>2,000 m²), small deposits (100 – 2,000 m²), and very

small deposits (< 100 m²; Bühler et al., 2009). Every avalanche has a release zone – a part where the avalanche is triggered, a track, a part where the snow is transported down the slope – and a runout, or deposition, zone (Fig. 2).

## 2 Previous work

Most previous work devoted to mapping avalanches from optical remote sensing data has focused on delineating runout zones. The idea of using object-based image analysis (OBIA) for detecting avalanches has been used in conjunction with

brightness information from aerial images and local slope data taken from digital elevation models (DEM), and with numerical modelling (Bühler et al., 2009), whereas others have used only the spectral information of aerial and satellite images to detect snow avalanches (Lato et al., 2012). Bühler et al. (2009) proposed an approach for mapping snow-avalanche deposits from ADS40 20-cm aerial images, which they resampled to 1 m and then combined with 25-m elevation data. They used the numerical simulation tool RAMMS (Rapid Mass Movement Simulation; Christen et al., 2010) to exclude slopes

>35° from the runout calculation, as they assumed these slopes could not accumulate snow-avalanche debris. They also used spectral thresholds to exclude snow-free areas. To separate rough avalanche debris from surrounding smooth and undisturbed snow, they used the normalised difference angle index (NDAI), evaluated from nadir and backwards NIR bands. They computed the NDAI difference between neighbouring pixels with a grey-level co-occurrence matrix (GLCM), which represents the distribution of pixel values at a given offset, and found that the thresholding of an entropy measure evaluated

via the GLCM achieved the best separability of rough and smooth snow. Ski lifts and other objects characterised by a similar entropy were removed using OBIA. The estimated accuracy of this method in terms of the fraction of correctly-detected avalanche deposits was 94%, and the producer's accuracy was 87%. Lato et al. (2012) applied OBIA for detecting avalanche deposits from panchromatic images only. They tested their algorithm with Quick Bird images in Norway, and aerial ADS40



images in Switzerland, relying on six variables (i.e. GLCM entropy, GLCM dissimilarity, brightness, contrast, similarity, and neighbour distance) in their procedure, in which segments failing to meet the OBIA assumptions were sequentially discarded. They started by eliminating dark regions from brightness data before detecting rough snow with edge contrast. The similarity filter and density helped to remove isolated pixels and small objects, respectively. Finally, the neighbour

distance helped to fill gaps inside the extracted snow-avalanche deposits. The user's and producer's accuracies of this classification were both >90%. Both these studies (Bühler et al., 2009; Lato et al., 2012) regarded OBIA as suitable for detecting snow-avalanche deposits, because it considers the spatial relation of the analysed segments in addition to their spectral properties.

Larsen et al. (2013) suggested an approach for optical Quick Bird imagery using directional filters evaluated based on image

texture classification (Varma and Zisserman, 2004) to distinguish avalanches from another objects. They assumed that avalanches have a texture pattern with a linear structure on the snow that coincides with the local hillslope aspect. Similar to other strategies, their classification took into account neighbouring pixels, while parameters such as area, area perimeter ratio, aspect direction difference, co-occurrence mean, correlation, and entropy were used to assist in excluding misclassified instances. Based on a visual comparison, the authors concluded that their classification was acceptable, allowing the

detection of many fresh avalanches with a low number of false alarms (Larsen et al., 2013). They pointed out, however, that some of the detected avalanches were split into parts; they therefore recommended additional processing to re-connect those fragments.

Eckerstorfer and Malnes (2015) manually detected avalanche debris based on its higher backscatter contrast, compared to the surrounding undisturbed snow cover, in Radarsat-2 Ultrafine SAR imagery. They assumed that avalanches are tongue-

shaped features with high surface roughness and higher snow density than surrounding terrain. Surface roughness and snow density were determined from backscatter, which increased in cases of higher surface roughness, and absorption, which increased for denser snow, respectively. They found that release zones and tracks were mostly difficult to detect. In a similar context, the automatic method of Vickers et al. (2016) evaluates backscatter in 50 x 50 pixel regions of Sentinel-1A images, subsequently masking out areas with a predicted probability of snow-avalanche occurrence of zero; pixels with a DEM-

derived local slope of >35° were removed from the occurrence of avalanches. From the test pixels, they selected those with a backscatter difference above a specified threshold. Randomly-selected pairs of pixels gave a total dissimilarity of pixels and class representatives for a K-mean clustering with two classes, 'avalanche' or 'not avalanche'. The detection rate (producer's accuracy) of this algorithm was 60%, and the authors highlighted its potential for avalanche monitoring despite masking out large amounts of data. Finally, Bühler et al. (2016) tested an unmanned aerial vehicle (UAV) that allows for fast, repeatable,

flexible, and cost-efficient measurements of snow depths in alpine terrain, possibly generating digital surface models of homogenous snow surfaces (Bühler et al., 2017). Legal regulations in Switzerland and elsewhere currently limit broad coverage of UAV imagery, however.





## 3 Study area and data

Our study area is centred around Davos, in the Swiss canton of Grisons; the area has alpine relief, with the highest local peak at Schwarzhorn (3,146 m a.s.l.; Fig. 3). Many slopes in this area exceed 28° and have dominant northeastern and southwestern aspects (Fig. 3). We used 0.25-m resolution NIR aerial images in conjunction with abundant avalanche

information acquired via a ADS80-SH92 large-format Digital Pushbroom Sensor (Leica Geosystems AG, Heerbrugg, Switzerland; Bühler et al., 2009) at the end of the 2012/13 winter season. The sensor recorded information with five spectral bands: panchromatic, blue, green, red, and NIR (Bühler et al., 2009). We have data for the area for more than five time slices starting from the winter of 2007/08 and continuing from 2011/12 to 2015/16. The images we used were taken at the end of the winter of 2012/13, where the highest expected snow depths were between 2,000 and 3,000 m a.s.l., and covered ~226

km².

## 4 Methods

We have introduced an automatic method for mapping release zones, tracks, and runout zones of avalanches using NIR 0.25-m aerial images. We compared the automatic classification with manually-digitised reference data and assessed the accuracy of detecting snow avalanches using confusion matrices. Furthermore, we verified the topographical conditions on which

most mapped avalanches occurred and verified two approaches for visualising the avalanche density. In addition, we proposed a probability approach to representing release and runout zones of avalanches and the automatic classification of snow avalanche parts.

### 4.1 Automatic OBIA snow avalanche classification

We implemented a multi-step OBIA approach for detecting avalanches in eCognition Developer 9.1.1 software (Fig. 4). As

input for the classification, we used the green, red, and NIR bands, and computed from these the normalised difference vegetation index (NDVI = $\rho_{NIR} - \rho_{Red} / \rho_{NIR} + \rho_{Red}$; Townshend and Justice, 1986), and the normalised difference water index (NDWI = $\rho_{Green} - \rho_{NIR} / \rho_{Green} + \rho_{NIR}$; McFeeters, 1996) and its standard deviation in a 5 x 5 kernel (SD$_{NDWI}$). We derived brightness as the mean of the green, red, and NIR bands to classify 'dark objects', such as rivers, rocks, and buildings. The NDVI helped to classify trees, bushes, and other types of vegetation, whereas the NDWI (SD$_{NDWI}$) detected snow (rough

snow; Fig. 4). We stretched computed NDVI and NDWI data into an interval of [0, 255].

### 4.1.1 Classifying vegetation, dark objects, and snow

In our first step, we segmented the data using a chessboard segmentation algorithm in eCognition 9.1.1 software, assigning a standalone segment to each single pixel (Fig. 4). We used the pixel values of snow-free areas obtained from NDVI and brightness to classify 'vegetation' and 'dark objects'. We classified vegetation as having positive NDVI values

(corresponding to >127 in the stretched range), and dark objects as those with a brightness of <4,000 (Fig. 4). Because pixels





on the border of vegetation and dark objects have mixed values, we assigned them to a separate 'buffer' class by reclassifying every pixel that shared a border with either 'vegetation' or 'dark objects' (existence of 'vegetation' or existence of 'dark objects'; Fig. 4), and then excluded this 'buffer' class from further analysis. We merged all segments classified as 'vegetation' and 'dark objects' and assigned all segment areas <6.25 m² (<100 pixels; Fig. 4) as being too small to buffer or

divert an avalanche, and included these segments as potential areas where a snow avalanche could occur. All the size thresholds used in our OBIA workflow were set to the resolution of the data that were used and the size of the analysed avalanches. Similarly, we classified pixels with a positive NDWI (>127 for stretched data) as snow and pixels with an additional roughness contrast ($SD_{NDWI} >1$) as 'rough snow', representing avalanches.

### 4.1.2 Removing small objects that do not represent avalanches

We then merged all snow pixels and reclassified 'rough snow' segments <12.5 m² (< 200 pixels, Fig. 4). We thus included many pixels as parts of avalanche deposits that had escaped being classified as 'rough snow' in the previous step because of $SD_{NDWI}$ values that were too low. We applied an assumption concerning the maximum area of segment that can be reclassified into the 'rough snow' class to avoid the inclusion of large, but smooth, areas inside avalanches. After comparing the segment values with their visual representation in an image, we observed that larger smooth areas that are inside the

avalanche represent small ascents that were omitted by the avalanche, and should therefore not be assigned as a part of an avalanche. We similarly assumed that the boundary of 'rough snow' should be equal to 1 for both 'snow' and 'buffer' classes, which means that the segment lies completely within these two classes; however, this time we combined it with a $SD_{NDWI}$ of >=0.75 to include the segment as 'rough snow'. The thresholding value of the $SD_{NDWI}$ was taken from the data histogram by analysing the rapid change in the counts of values on the histogram. We than merged the segments into the

'rough snow' class, with the exception of areas <62.5 m² (area <1,000 pixels; Fig. 4), assuming that they were too small to represent an avalanche.

### 4.1.3 Buffering

In further steps, we split all segments from the 'rough snow' class into smaller pieces, to reduce their artificially-complex shapes with bigger and more compact parts connected to neighbours by only a few pixels. In most cases, only some of these

complex shapes represented an avalanche, whereas the remainder was 'rougher snow' due to vegetation effects. To simplify these shapes into separate parts, we used buffering to reclassify pixels from the 'rough snow' class as snow that had less than four neighbours classified as 'rough snow' (Fig. 4), repeating this step for both the 'snow' and 'rough snow' classes. To avoid undue growth of spurious pixels, we narrowed down the process to only pixels adjacent to at least one pixel classified as 'rough snow' (Fig. 4).





### 4.1.4 Neighbourhood analysis

At this stage, our classification still contained many misclassified parts of avalanches containing effects of vegetation, soil, or rocks. To assign these parts to the 'rough snow' class, we first reclassified all pixels from 'unclassified', 'buffer', 'vegetation', and 'dark objects' with brightness >3,000 and NDVI <140 into a new 'temp' class (Fig. 4). After merging the
segments into the 'temp' class, we reclassified all segments bordering 'rough snow' <0.01 into 'unclassified' and discarded these from the analysis. For the remaining segments, after returning to chessboard segmentation to allow us to once again operate on single pixels, we reclassified all pixels from the 'temp' class sharing a border with 'rough snow' with at least two pixels (Fig. 4) iteratively. Because every pixel had only four neighbours, the assumption concerning two pixel neighbours stopped the infinite loop after only a few repetitions. To allow the inclusion of additional pixels, we therefore decreased the
threshold on the relative border to >=0.25 and performed the process twice more to allow more segments that were sharing the boundary with at least one 'rough snow' segment to be included in the 'rough snow' class. We did not run this process iteratively, because it would have reclassified all pixels assigned as 'temp', and our aim was to increase only compact 'rough snow' areas. Next, we once more applied an infinite loop regarding the relative border to 'rough snow' of >=0.5, to increase previously-detected snow avalanches. These steps were crucial in closing areas inside the snow avalanches that due to the
values of brightness and NDVI, were not assigned in the previous steps to the 'rough snow' class.

### 4.1.5 Adding small gaps inside avalanches

Finally, we focused on filling gaps inside the detected snow avalanches. We reclassified and merged gaps into avalanches by verifying their geometrical relation to 'rough snow'. After checking the layer statistics for every segment, we built an assumption that if a segment was completely within the 'rough snow' and its area was <1,000 pixels (62.5 m²), it was to be
automatically reclassified into 'rough snow' (relative border to 'rough snow' = 1 and area <1,000 pixels; Fig. 4). Segments with >1,000 pixels reclassified into 'rough snow' were expected to fulfil additional rules concerning their roughness, brightness, occurrence of snow, and vegetation, because they may have represented a convex-upward form that could stay intact during the avalanche occurrence. Only segments with a high snow roughness that did not represented vegetation and were not too dark were added to the 'rough snow' class (relative border to 'rough snow' = 1, $SD_{NDWI}$ >0.7, NDWI >127,
NDVI <140, and brightness >2,500; Fig. 4). We exported all extracted snow avalanches into polygon shapefiles and compared these visually and quantitatively with manually-mapped reference data. The visual interpretation was important for verifying the distribution of errors and the completeness of classified avalanches.

### 4.2 Generating reference data

We created reference data by manually digitising avalanches from the images in ArcMap 10.3 software at scales between
1:800 and 1:1,500, depending on the complexity of the mapped avalanche. Manually digitising each area affected by the avalanche was necessary because avalanches occur only during the winter season when snow cover occurs, making them



temporal events; the marks of their existence disappear when the snow cover melts, and therefore no complete reference data are available. Avalanches that were overlapping or bordering others were counted as one. This means that polygons representing an avalanche in fact covered several smaller avalanches that had occurred in succession; in most cases, it was impossible to assess their relative sequence (Fig. 3). We mapped a total of 2,200 avalanche polygons for the data acquired in the winter of 2012/13, obtaining 13.6 km² of avalanche terrain or 6% of the study area. The reference data, as well as automatically-extracted avalanche polygons, are available online ([https://uni-potsdam.maps.arcgis.com/apps/Cascade/index.html?appid=3b5ac4491b59480c8c6016139f285e88](https://uni-potsdam.maps.arcgis.com/apps/Cascade/index.html?appid=3b5ac4491b59480c8c6016139f285e88)). We used the reference data to estimate several classification accuracy metrics, including Type I, Type II, and total errors (Sithole and Vosselman, 2004), overall, user's, and producer's accuracies (Congalton, 1991), Cohen's kappa (Cohen, 1960), and F-Score. Finally, we arbitrarily selected three 4-km² training sites (Fig. 3) for our OBIA algorithm, and reported these eight performance metrics for a larger 226.3 km² test area.

## 5 Results

### 5.1 Estimated accuracy

Our algorithm classified 10.7 km² as avalanche debris, which is 78.7% of the total area of the reference data mapped for the winter of 2012/13. Overall, 1,648 out of 2,200 avalanches were correctly identified; 1,126 were detected in terms of more than half their area, and for 615 avalanches this detection rate was >80%. These classified avalanches were spread out evenly throughout the study area (Fig. 5). Visual checks of the classification indicated that the runout zones were detected most reliably, whereas the release zones were the most problematic. Tracks were detected mostly correctly where small patches of vegetation or soil were near or in the avalanche tracks. The highest estimated precision in detecting avalanche boundaries was in runout zones adjacent to smooth snow or unvegetated slopes. In some locations, a clear distinction between avalanche debris and smooth snow was not possible, especially for older deposits or snow drifts. Fresh avalanches were also detected with higher accuracy than were older and blurred ones.

The performance metrics estimated for our training sites were overall and user's accuracies of >0.9, and a Cohen's kappa of >0.8 (Tab. 1). The accuracy for the testing area yielded lower performance metrics, with a user's accuracy of 0.78. The producer's accuracy showed that 61% of the total avalanche area in the tested data was correctly identified. The overall Type II error was very low, indicating that few objects were falsely classified as avalanches, whereas the high Type I error showed that many, mostly old, snow avalanches remained undetected.

**Table 1: Performance metrics estimated for 4-km² training sites 1 – 3 (see Figures 3 and 5) and for the entire study area covering 226.3 km².**

| | Type I error | Type II error | Total error | Overall accuracy | Producer's accuracy | User's accuracy | Cohen's kappa | F-score |
|---|---|---|---|---|---|---|---|---|





| | | | | | | | | |
|---|---|---|---|---|---|---|---|---|
| Site 1 | 0.23 | 0.01 | 0.05 | 0.95 | 0.77 | 0.91 | 0.81 | 0.83 |
| Site 2 | 0.23 | 0.02 | 0.08 | 0.92 | 0.77 | 0.92 | 0.79 | 0.84 |
| Site 3 | 0.16 | 0.02 | 0.05 | 0.95 | 0.84 | 0.93 | 0.85 | 0.88 |
| Total | 0.39 | 0.01 | 0.03 | 0.97 | 0.61 | **0.78** | **0.67** | 0.69 |

## 5.2 Influence of variables used for classification accuracy

We checked how brightness, NDVI, NDWI, and $SD_{NDWI}$ derivatives evaluated for each classified avalanche affected the producer's accuracy. For each avalanche, we computed the mean for each derivative map using the values of all the pixels inside the avalanche. We found that avalanches that were extracted with the highest accuracy were generally also brighter (Fig. 6); an increase in producer's accuracy occurred with an increase in the mean brightness of avalanche. A similar, but, weaker correlation held for $SD_{NDWI}$. Neither NDVI nor NDWI had much of an influence on the classification accuracy. In addition, we verified if an avalanche's shape (roundness) and size (area) affected the detection rate. Similarly to NDVI and NDWI, however, we did not find any dependence (Fig. 6).

## 5.3 Topographic factors favourable for snow avalanches

We further analysed the topographic settings of the mapped avalanches. Most avalanches (1,422 out of 2,200) occurred between 1,900 and 2,600 m a.s.l., with a mode of approximately 2,400 m a.s.l., on slopes that were 20 – 40° (Fig. 7). One hundred thirty nine out of 193 of the highest-lying avalanches (>2,800 m a.s.l.) were small or very small events, according to the nomenclature of Bühler et al. (2009) and occurred in the southern part of our research area. Although the largest avalanches occurred below 2,400 m a.s.l., most affected northeastern and the southwestern slopes, thus mimicking the major aspects of the mountain ranges (Fig. 3).

## 5.4 Density of avalanches

Using the reference data for the winter of 2012/13, we further analysed where most of the avalanches occurred. We computed centroid locations for each avalanche polygon, and estimated their spatial density using the *Kernel density* function in ArcGIS 10.3, with both point- and area-weighted inputs in a 2-km radius. We produced two maps because, due to the occurrence of multiple avalanches, our input centroids did not represent the total number of events. Consequently, a point-weighted map could have underestimated the real avalanche density, whereas an area-weighted map avoids this issue. The selected 2-km size of the bandwidth was large enough to avoid reproducing the pattern of input avalanche centroids, and small enough to reduce the smoothing of the point information. We found that avalanches clustered largely in the south-eastern part of the study area (125 smaller avalanches in inset 1, Fig. 8, of which 104 were <2,000 m²). The area-weighted spatial density of avalanches was highest on the slopes of Fluela Schwarzhorn, Sentischhorn, and Wuosthorn. The biggest avalanche in inset 2 on Figure 8 had an area of ~390,000 m².





### 5.5 Automatic classification of snow-avalanche zones

Automatic delineation of release and runout zones from remote sensing data can be conducted by verifying the elevation values in each polygon that represents an avalanche. The simplest approach uses a flow length, which represents the distance along the flow path inside the avalanche, as an indicator in evaluating the probability of the release and the runout zones.

Herein, we suggest using the elevation values in an approach that allows exposure of local terrain height differences in the probability map. For example, applying the elevation for a release zone that occurs on very steep slopes determines that the highest probability will be represented only for a very small area, whereas on gentle slopes, the same probability will occur over a larger area, because the elevation differences are smaller than those of steep slopes.

We used a 2-m resolution digital surface model (DSM) derived from stereomatching of aerial ADS images to automatically

detect release zones, tracks, and runout zones. Because we generated our reference data in the same resolution as the aerial images, we first resampled the DSM to 0.25-m resolution (Fig. 9). We then used the reference data to clip the DSM to each avalanche polygon using a key ID. DSMs acquired in this way were used to compute the probability of representing individual parts of an avalanche. We assumed that the maximum (minimum) elevation on which the snow avalanche occurs has the highest probability of being a release (runout) zone. We estimated the probability by normalising elevation data (Fig.

9) by subtracting the minimum value of DSM inside this avalanche from DSM pixel value, and dividing by the difference between the maximum and the minimum values inside each avalanche. Such stretching returns a $0 - 1$ probability map for release areas. The thresholds in classifying the track are relative; we applied $0.3 - 0.8$ as the thresholds, because most of our 2,200 manually-digitised polygons represented loose snow avalanches where the release zone constituted a small part, and the runout zone compared to the track constituted an even smaller part. When dealing only with loose avalanches, the upper

threshold for release areas may be set to a probability equal to 0.9. The selected thresholds worked well for single avalanches, where both the release zone and the runout zone were assigned correctly (Fig. 9). For avalanches with more complex shape and multiple avalanche arms (e.g. avalanche 1,207 on test site 2; Fig. 9), however, the release zones remain undetected, with the exception of only the highest-lying release areas.

### 6 Discussion

Automatic mapping of avalanches is crucial in mountainous regions to delineate susceptible areas and to produce cadastres for validating avalanche forecasting and hazard maps. Previous studies have demonstrated that a combination of aerial images and digital elevation models (Bühler et al., 2009), or aerial (Lato et al., 2012) and satellite (Larsen et al., 2013), or SAR images (Vickers et al., 2016) allow the detection of snow avalanche deposits; however, identifying release zones and tracks remains challenging. We have proposed an OBIA algorithm that tracks release zones, tracks, and runout zones in NIR

images. We recommend using normalised derivatives NDWI and NDVI, instead of brightness, for classifying water (snow) and vegetation, respectively, because the thresholds allowing for classification of water and vegetation in these indices are stable (around zero) and widely applicable. Our approach upon expands previous work, as we have considered potential





snow avalanches in snow-covered areas only (NDWI >127), whereas others have used brightness thresholds (Lato et al., 2012), with the danger of including other objects with similar values to those of snow. Rivers and lakes have similar values of NDWI, but are less bright than snow, we therefore combined NDWI and brightness in our model. We suggest using $SD_{NDWI}$ to trace rough snow or avalanche debris. Estimated accuracy was high for detecting avalanches using the test data, and the random spread of avalanches detected with high accuracy through the whole research area suggests that the assumptions in our approach are broad enough to be applied for a large area.

To increase the accuracy of true positives in detecting release zones in our classification, a customised threshold for every data tile can be applied. In any case, we focused on developing a method that is transferable and works well for a greater area. We also wanted to test the usability of the image derivatives and verify how much information we could obtain from them when detecting snow avalanches. We therefore did not change any parameter or its threshold when applying the algorithm for other data tiles (Lato et al. 2012). We implemented several steps that allowed the classification of different avalanche scenarios that occur in diverse topographic conditions, such as single vs. multiple avalanche; small vs. large avalanche; avalanche revealing the ground or vegetation vs. avalanche that does not reveal the ground or vegetation; avalanche that is blocked by vegetation or other objects having high roughness vs. avalanche that is not adjacent to any rough object. These topographic conditions influence the appearance of avalanches on images; therefore, the number of steps in our OBIA workflow is large. We found that OBIA is useful for complex shapes, because it allows the implementation of assumptions regarding each different situation. Additionally, this OBIA algorithm may be used and modified according to the need of the user and the data. For example, some steps in our OBIA workflow may be omitted, such as those used for filling the gaps inside the avalanche. If the data do not contain avalanches that reveal bare ground or vegetation, these steps may not improve the classification, because there will not be any gaps to reclassify. Additionally the number of loops in the shrinkage and growth steps depend on pixel size. With our 0.25-m resolution NIR images, we shrank and increased the segments of 1.25-m, which was sufficient to split segments into small parts; with higher (lower) resolution, the number of loops should be increased (decreased) accordingly. Previous studies have reported very high accuracy in separating vegetation and water using vegetation and water indices (Townshend and Justice, 1986; Ji et al., 2009); therefore, we assume that the NDVI, and NDWI thresholds are stable and may be easily transferable to other areas and data. Despite this, the thresholds for $SD_{NDWI}$ should always be verified by analysing the mode of data distribution on a histogram of the $SD_{NDWI}$, testing the thresholding of data samples, and checking the thresholding results visually. We are aware that such visual checking may introduce some bias, although so far it is the most common way in OBIA to find the most suitable threshold.

The size and the shape of avalanches did not influence the classification accuracy if they were bigger than 2,000 pixels, which is what we regard as the minimum detectable size in our OBIA algorithm. We selected the minimum number of pixels necessary by analysing the reference data, where avalanches smaller than 2,000 pixels constituted only 7.5% of the total number of avalanches, and by taking into account the classification of avalanches with respect to their size. According to Bühler et al. (2009), very small avalanches are <100 m², which coincides with 2,000 pixels (125 m²). The parameters playing the biggest roles were brightness and the $SD_{NDWI}$ (Fig. 6). Visual inspection showed that the easiest avalanche part to detect



was the runout zone and the most difficult was the release zone, because release zones were not usually rough enough or did contain outcrops of the vegetation or bare ground. The correct detection of the track depended mostly on snow roughness and depth. In a case of low roughness values or a very thin snow cover revealing the ground and the vegetation, the track was not detected correctly or not detected at all.

The most difficult to classify were old avalanches where the bare ground cropped out or where vegetation occurred in the path of the avalanche. These avalanches did not meet the assumptions in our OBIA protocol and could not be classified correctly because they were not rough enough, or were too vegetated or dark due to thin snow cover. We tested different thresholds for the input layers and different neighbourhood assumptions in order to include these avalanches; however, this resulted in more false positives, so the cost of correct classification of these avalanches was higher than the benefit. We

therefore decided to stay with the same workflow and thresholding shown in Figure 4 for the whole study area.

Errors in automatic classification also occurred due to data tiling. Using an Intel Xeon E5-2667U processor with 256GB RAM memory, we were able to run our OBIA algorithm only for tiles of 6.25 km² (10,000 columns x 10,000 rows), requiring a computing time of ~30 minutes for each tile; our test in executing bigger tiles ended in crashing the computation in the eCognition software. In many cases the avalanches were therefore split into two or more neighbouring tiles, which

influenced the correct detection of avalanche parts, especially those where the avalanches were spread across tiles or had small gaps.

Avalanches often have a tongue shape that may also be used as a property for classifying. In our test area, however, using such information may be insufficient, because we are dealing with avalanches repeated in the same location and merged avalanches, which have more complex shapes. In such cases, a new avalanche may partly cover previous avalanche, making

it difficult to distinguish them. In addition, a few avalanches may simply have a more complex and difficult-to-interpret shape.

Our estimated avalanche density map and information concerning the most common prevalence of avalanches may be used to help generate a hazard map in mountainous areas. The most crucial issue, however, is whether the size or the number of avalanches is more important in such mapping. As mentioned by Eckerstorfer et al. (2016), only very small avalanches cause

less damage; therefore, we suggest weighting the density map according to avalanche size to give more information regarding the degree of danger in a specific area.

Continuation of our research should contain verification of the transferability of our OBIA algorithm to data from different winter years for the area of Davos, Switzerland, which we have in our repository. In addition, future tests involving diverse areas across the World and different types of data (e.g. UAV) are desirable. An approach for distinguishing single and repeat

avalanches at the same area should be developed, because it would give more detail about the quantity and the frequency of avalanches in a given area.





## 7 Conclusions

We have presented an automatic object-based image analysis (OBIA) approach to detecting snow avalanches and their release, track, and deposition zones for a large region from ADS80 NIR aerial images. We used image-derived parameters, including the normalised difference vegetation index (NDVI), and the normalised difference water index (NDWI) and its

standard deviation ($SD_{NDWI}$), to separate vegetation, snow, and rough snow representing avalanche debris, respectively. We applied buffering assumptions relying on the local neighbourhood of segments to remove salt-and-pepper noise and objects that were falsely assigned as avalanches, using the thresholding of derivatives obtained from NIR images. For an area of 226.3 km², our algorithm achieved producer's and user's accuracies of 0.61 and 0.78, respectively, and a Cohen's kappa of 0.67. Our algorithm uses only information taken from these images, from which the evaluated NDVI and NDWI indices are

normalised, helping to transfer their thresholds to other areas. Our approach contains only three fixed parameters (NDVI, NDWI, and $SD_{NDWI}$) and two changeable parameters (brightness and segment area). The first depend on the spectral characteristics of images and the latter one depend on the resolution of images. Our OBIA workflow is not sequential when compared to that of Lato et al. (2012); the segments that do not fulfil assumptions in one step can still be considered as potential snow avalanches in the next steps. To assign potential snow avalanches, our method takes into account only the

rough snow, which allows more reliable detection of avalanches. Our probability approach determines, in an automatic way, the highest and the lowest parts of the avalanche, and thus its release zone, track, and runout zone, which allows easy analysis of the topographic condition of areas where the avalanche starts and where the snow is deposited. For multiple avalanches with a complex shape, our probability map may not be sufficient to correctly identify all release and runout zones, but for single avalanches, it gives valuable results. For some avalanches, we were not able to judge visually if they

were single or multiple; a discussion of this topic should therefore be undertaken. The probability approach may be used for any other mass movement landforms, such as landslides, to delineate their release and deposition zones. In the future, we plan to validate our snow avalanche algorithm for ADS data, which we have for other winters, and to verify its transferability to other NIR images, because successful results in this matter may offer a chance to improve hazard maps and avalanche forecasting in Switzerland.

**Acknowledgements**

This research was funded by the European Union under the Marie Curie Initial Training Network ALErT (Creation of an interactive CAP natural-hazard database), project-number: FP7-PEOPLE-2013-ITN-607996. The ADS80-SH92 airborne images used in the study were provided by the WSL- Institut für Schnee- und Lawinenforschung SLF, Davos, and Leica Geosystems AG (R. Wagner, N. Lämmer, F. Schapira). The authors would like to thank J. Wessels and B. Zweifel for

providing historical snow avalanche data for Switzerland.





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





**Figure 1: Snow avalanche accidents with victims in Switzerland in the winters of 1996/97 to 2015/16, and the Alps percentage per country. Data from: Swiss Federal Institute for Snow and Avalanche Research (WSL-SLF), Davos, Switzerland.**





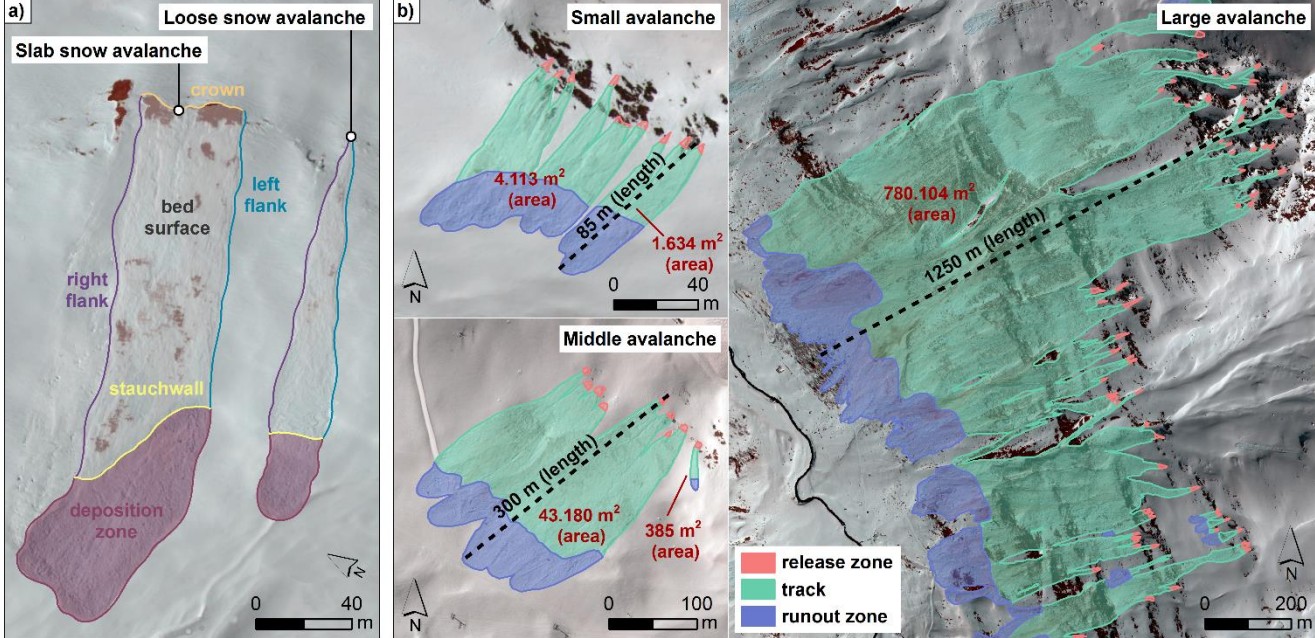

**Figure 2: a) Types of avalanche release: slab avalanches and loose avalanches, with marked avalanche body parts (crown, bed surface, stauchwall, deposition zone, and right and left flanks; Schweizer et al. 2003); b) Avalanche classification with respect to length: small, middle, and large [m], with the avalanche area [m²] and marked parts of avalanches: release zone, track, and runout zone. Data from: WSL Institute for Snow and Avalanche Research (SLF), Davos, Switzerland.**




**Figure 3: Topographic setting of research area and ADS 80 NIR aerial images with test sites (1 – 3) and digitally mapped snow avalanches; different colours mark different snow avalanche polygons. Inset histograms show a) distribution of local slope and b) the main slope aspect.**





**Figure 4: Workflow for classifying snow avalanches with object-based image analysis (OBIA). The white boxes indicate the classification; dashed outlines feature the number of reiterations (red), and the local decision boundaries show change (blue). The colours of squares are coded to the input and output class in each step. Figs. a), b), c), and d) represent visual results of the sub-step classification.**



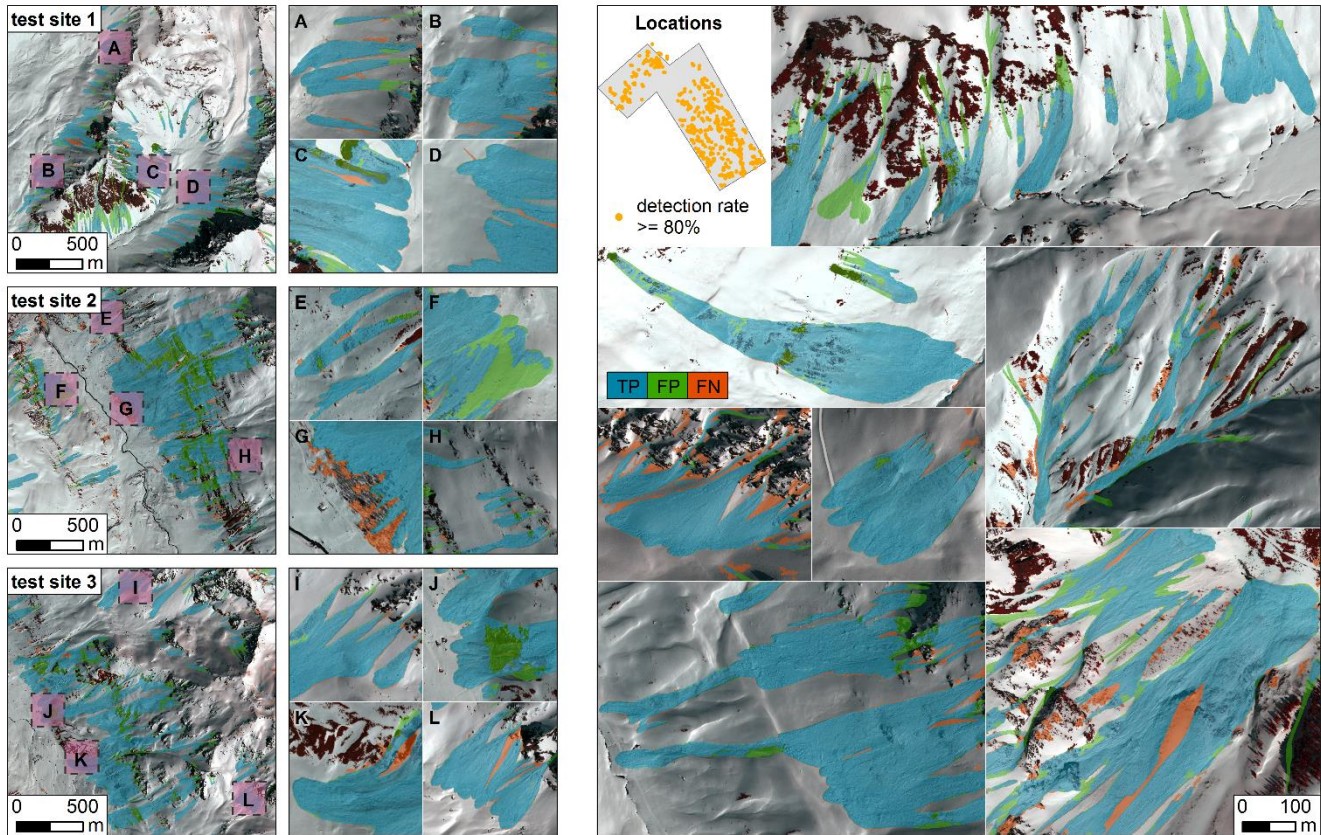

**Figure 5: Accuracy assessment of OBIA classification for training (test sites 1 – 3) and testing data together; avalanche debris are shaded with orange colour, where they were detected with an accuracy >=80%. TP = true positive; FP = false positive; FN = false negative.**





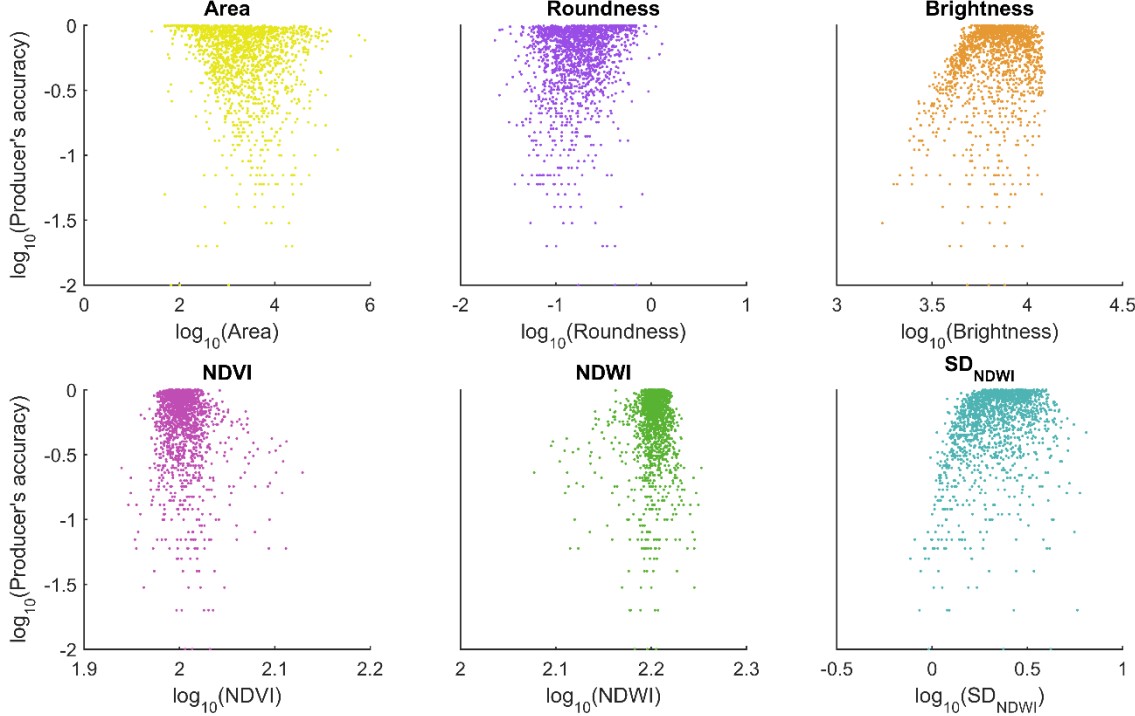

**Figure 6: The role of avalanche area, roundness, brightness, normalised difference vegetation index (NDVI), normalised difference water index (NDWI), and standard deviation of normalised difference water index (SD$_{NDWI}$) in the estimated accuracy, when detecting snow avalanches with our OBIA approach.**





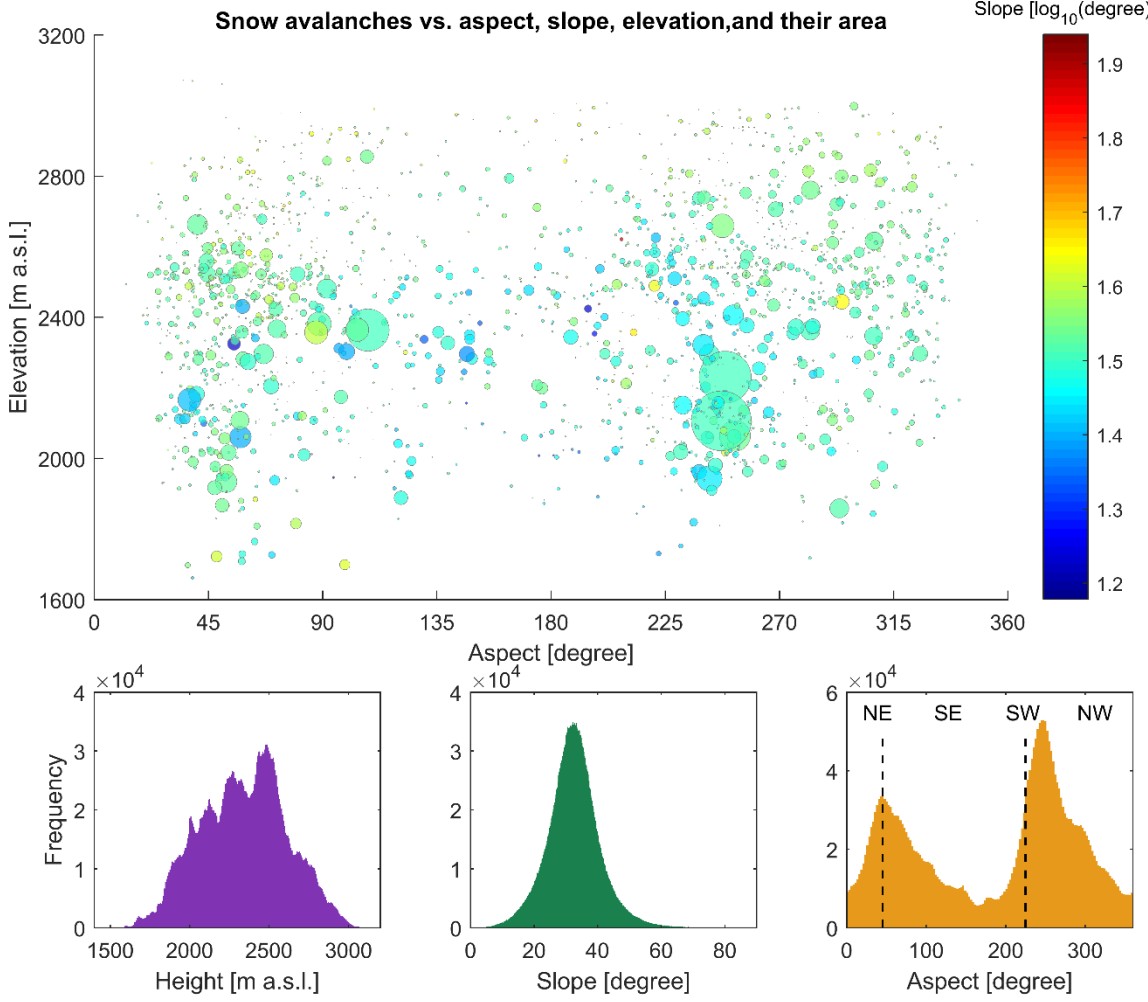

**Figure 7: Elevation, slope, and aspect of mapped reference avalanches with marked northeast and southwest directions in the winter of 2012/13. The values presented for every single avalanche represent the mean value of each pixel contained inside the reference avalanche polygon. Bubble size on the scatter plot is scaled to avalanche area.**





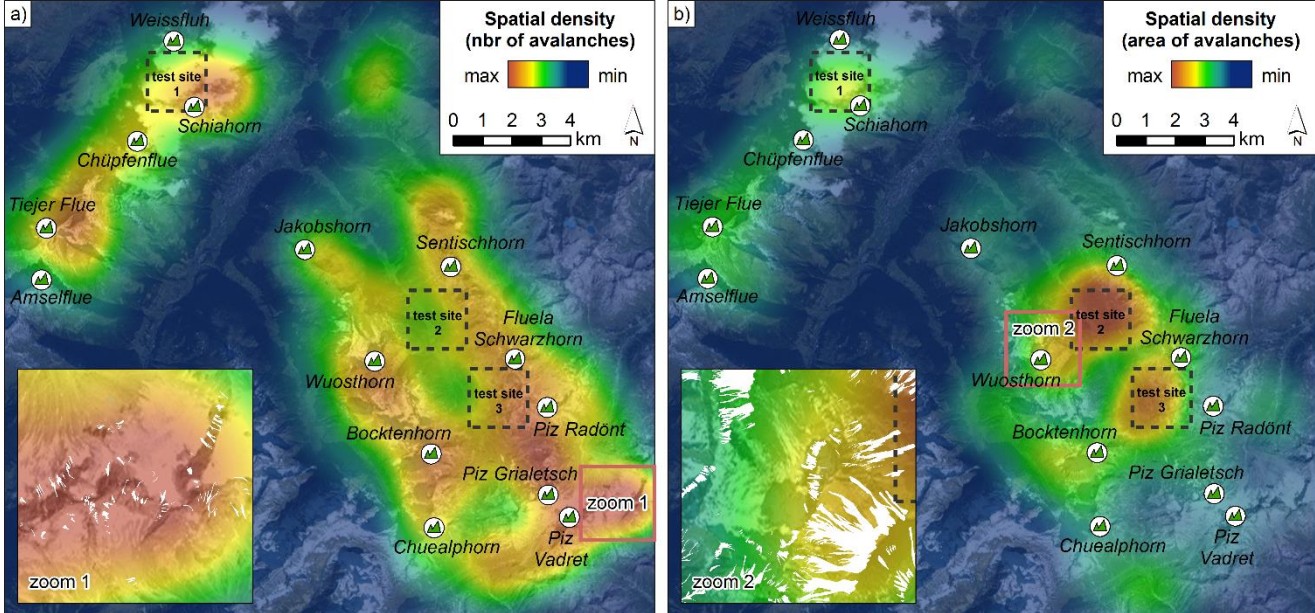

Satellite image source: ESRI

**Figure 8: Density maps showing the clustering of avalanches in the winter of 2012/13, with respect to their a) quantity, and b) size. The insets (zoom 1 and zoom 2) show the density maps with respect to manually-classified reference snow avalanche polygons shown in white. Outlined test sites 1 – 3 are the test sites that were used for developing our OBIA algorithm.**



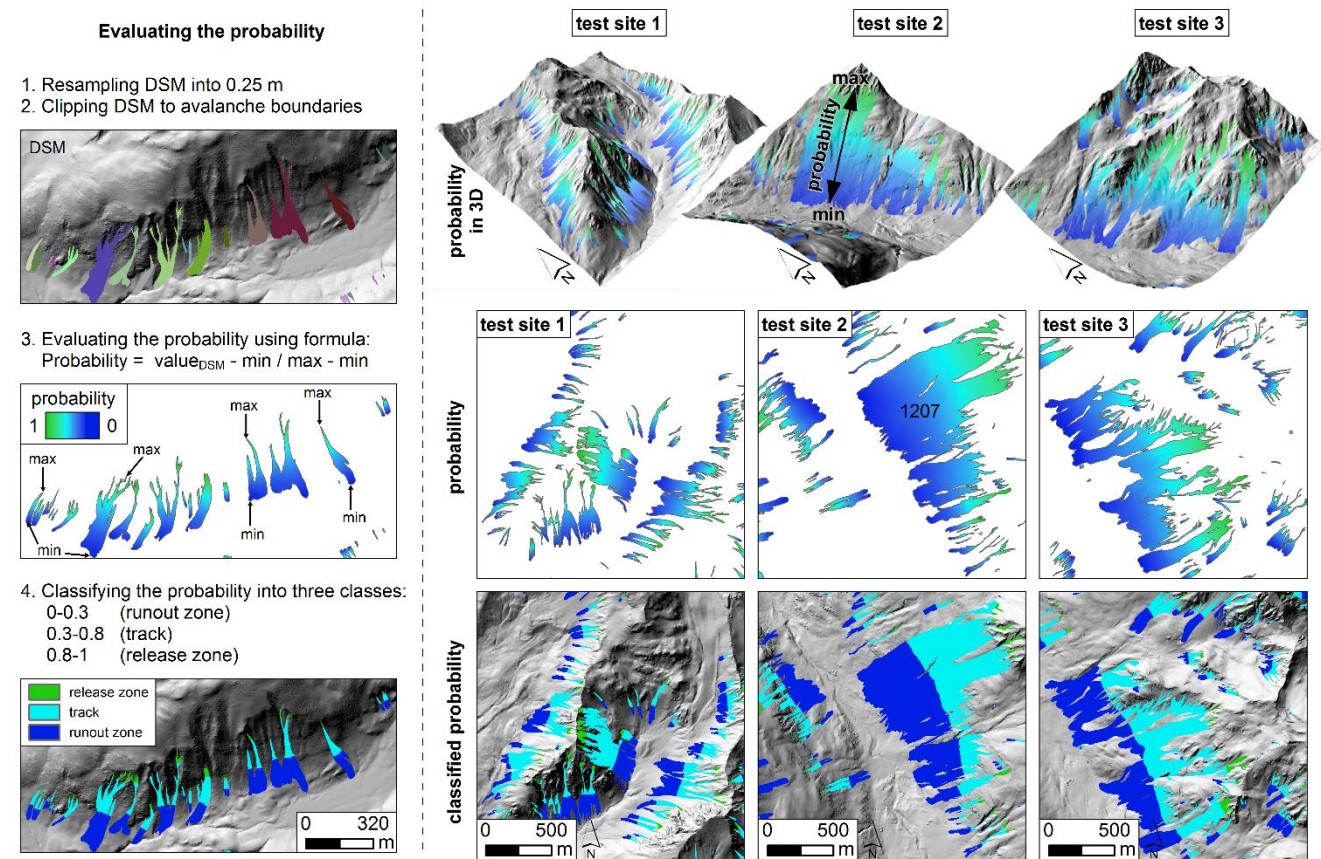

**Figure 9: Estimating the probability of an avalanche release area. A probability close to one indicates a pixel representing a release zone, whereas a probability close to zero indicates a runout zone in automatically classified data.**