# Peer review of "Regional snow-avalanche detection using object-based image analysis of near-infrared aerial imagery"

_Natural Hazards and Earth System Sciences, 2017_

## Referee Comment (RC1) · Anonymous Referee #1 · 24 May 2017

General comments:

The paper by K. Korzeniowska and co-workers describes a method for automatic avalanche detection based on object-based image analysis (OBIA) of near-infrared (NIR) aerial imagery. The paper starts with an introduction of the dangers and risks related to snow avalanches with some detailed information on human fatalities and damages to infrastructures (mostly focused on the case of Switzerland), and the need for further developments concerning the large-scale yet precise mapping of avalanche activity. Section 2 provides the reader with an overview of the existing studies on automatic detection of avalanche activity based on image processing. A very short section (current section 3) describes the study area and data, and already includes

some information on the methods. Section 4 is the detailed section on the methods. Section 5 shows the main results with a focus on the following points: the accuracy of the OBIA workflow proposed, including the influence of variables chosen -brightness, NDVI, NDWI, SD_NDWI- on accuracy, on the one hand, and the results on (i) topographic factors that influence avalanche activity, (ii) density of avalanches and (iii) classification in release, flow and run-out zone of detected avalanches on the other hand. Section 6 is a discussion on the advantages and drawbacks (points to be improved) regarding the method proposed. Finally, the last section concludes by mostly giving a summary of the main points discussed in section 6.

The paper presents an interesting method for detection of avalanche activity, which is based on object-based image analysis (OBIA) of near-infrared (NIR) aerial imagery. Given that NIR aerial images of good resolution are available, this method has the potential to cover large-scale mountainous areas including very remote areas. Furthermore, a statistical approach allows to distinguish between release, flow and run-out zones of the detected avalanches. One tricky yet important issue remains the distinction between single and multiple events. The OBIA workflow and the results presented in the paper are worth to be published in NHESS. However, the paper in its current form is not ready for publication. Some substantial revision is needed at least for two main reasons. First, I found that the sensitivity analysis of the method and outcomes to varying the different thresholds is lacking. The authors should pay attention to including such a sensitivity analysis when possible, and/or argue more on their choice regarding the thresholds of a number of parameters: see my specific comments below. Second, the authors should make an effort to improve the editing/structure of their manuscript that is sometimes quite hard to follow: short section versus another much longer section, announcement of outlines needed in the main introduction and the long section on "methods", etc. (see my comments regarding Editing/Structure of the paper and technical corrections).

Specific comments:

- section 1 (intro), page 2 (lines 20-21): Saying "..., focussing mainly on geographic coordinates, but rarely on any detailed information about their extent or area." appears too strong to me. I think the authors should qualify that statement. Traditional methods, such as photo-interpretation, the use of testimonies, photographs, etc, gives crucial information, and merging that information into one single platform (see for instance the paper by Irstea group, by Bourova et al. CRST 2016, as well as some references therein) is already an important and efficient step. Could you please revise this part of the text?

- section 2, page 3 (lines 23-25): Please could you argue a little bit on this 35 deg threshold? I believe such a limit should depend on the snow type and then influence the results of your automatic detection approach. Please could you comment on this point?

- section 2, page 4 (line 25): again, this threshold of 35 deg needs much more discussion. This angle should depend on the snow type. Either you show a sensitivity analysis of your detection to that threshold, or you give more physical arguments on this choice.

- section 4.1.1, page 6 (line 4): why this threshold of 6.25 square meters (the exponent 2 for the segment area is missing)? Could you please argue to make your choice less arbitrary? Did you conduct any sensitivity analysis of your method to varying that threshold? If yes, could you please include a thorough discussion on that analysis in the manuscript?

- section 4.1.2, page 6 (lines 13-16): the sentence is not clear to me, please revise. Could you please show a couple of examples of the cross-comparison between segment values and their visual representation in an image?

- end of section 4.1.2: could you please argue on this threshold of 62.2 square meters? Again: did you conduct any sensitivity analysis? Also I would suggest you show a square of that size on one figure (among figs a,b,c,d shown in Fig. 4 for instance). It

would be useful for the reader to materialize the physical size of that threshold onto the map, compared to avalanche extensions that are detected.

- section 4.1.4, page 7 (line 4): could you please explain why you are using/choosing those values for both brightness and NDVI thresholds? I would suggest you to show a sensitivity analysis to varying those thresholds. This seems to me a crucial point if you would like your method to be possibly extended to much larger scales or other mountainous areas.

- section 4.1.5, page 7 (lines 24-25 in brackets): why those thresholds? Could you show a sensitivity analysis to varying the thresholds? Brightness threshold is 2500 here, while it was 3000 a little earlier in the text (see previous comment too).

- section 5, page 8 (lines 17-22): this part discusses the problems/limits of the method. The reader would like to know how those problems/limits are sensitive to the choice of the different thresholds (see the specific comments above). Could you please strengthen the discussion on this point?

- section 5.3, page 9 (line 12): is the range 20-40 deg compatible with the threshold of 35 deg discussed in section 2?

Editing/Structure of the paper and Technical corrections (typings errors, etc.):

- Abstract (line 19): "... at three 4-km areas..." The exponent 2 for areas in square kilometres per square is missing?

- end of section 1 (intro), page 3 (lines 14-15): again, the exponent 2 is missing here. Please fix it.

- section 1: would be nice to terminate the section by announcing the detailed structure of the paper, with a short summary of each section (including explicit numbered reference to each section).

- section 2, page 3 (lines 23-25): the construction of the sentence is a bit weird... The

threshold of 35 deg is an assumption (a model input that stems from the DEM data) but not an outcome of the RAMMS model. Please revise the sentence.

- section 2, page 3 (line 27): "nadir?". Please fix this.

- section 3, page 5 (line 10): again, the exponent 2 is missing... PLEASE CHECK CAREFULLY the entire text regarding this issue, which is present in many parts of the manuscript.

- section 3 is very very short. Given the fact that it already includes some information on the methods, I would suggest that the authors include section 3 as a sub-section of section 4 "Methods".

- introduction of section 4: this part needs careful revision. Would be nice to put here an outline with explicit references to the sub-sections that follow (by using the numbering), as well as explicit reference to key figure 4. In its current form, I must say that section 4 is very difficult to read/follow.

- end of section 4.1.2, page 6 (line 19): typo... should be "then" instead of "than"

- section 5.4: I would suggest to replace "inset 1, Fig. 8" by "inset on Fig. 8a" and "inset 2 on Figure 8" by "inset on Fig 8b".

---

## Referee Comment (RC2) · O. Jaquet (Referee) · 8 Aug 2017

This paper presents an automatic object-based image analysis approach for the detection of snow avalanches using specific remote sensing data. The approach is explained in details and some statistical tests are performed for the evaluation of results uncertainty in relation to a reference data set. Following additional explanations/ comments would be valuable for the reader : I. A description of the conceptual model associated to the OBIA approach, it would provide some insight in relation to uncertainties and limitations; since only 61% of the total avalanche area was correctly identified... II. Mathematical details of the selected probability approach; to my understanding an

uniform distribution is selected in relation to the elevation ; the selection of such simple statistical model should be justified in relation to point I. III. A reference for confusion matrices IV. The impact of snow thickness on the results V. The potential use of additional data (radar, seismic, ...) to improve the results

---

## Author Comment (AC1) · 10 Aug 2017

Department of 3D Mapping
BSF Swissphoto GmbH
Mittelstrasse 7
12529 Schönefeld, Germany

Schönefeld, 10th August 2017

Dear Editor,

We are pleased to submit our answers to the Reviewer's comments of nhess-2017-120 "Regional snow-avalanche detection using object-based image analysis of near-infrared aerial imagery" article.

We would like to thank two reviewers for their constructive criticism and comments, and appreciate theirs insightful analysis of our study. We have considered all comments to improve the quality of the article. Following the suggestions, we restructure all sentences which might be misleading for the reader, we also add more description on sensitivity analyses and the thresholds we have selected in our approach. We will add more description on limits and problems that occur during the classification process. In addition, we will expand the limitations and the conceptual model of the OBIA approach in the discussion section of our manuscript. We will also correct the figures according to reviewer's request, and verify the text to avoid the typos.

For the convenience, we address each of the reviewers' concerns as outlined below:
- Reviewer's comments (**bold**),
- Our answers (*italic*).

Yours sincerely,

Karolina Korzeniowska
(on behalf of co-authors)

Anonymous Referee #1

General comments:

The paper by K. Korzeniowska and co-workers describes a method for automatic avalanche detection based on object-based image analysis (OBIA) of near-infrared (NIR) aerial imagery. The paper starts with an introduction of the dangers and risks related to snow avalanches with some detailed information on human fatalities and damages to infrastructures (mostly focused on the case of Switzerland), and the need for further developments concerning the large-scale yet precise mapping of avalanche activity. Section 2 provides the reader with an overview of the existing studies on automatic detection of avalanche activity based on image processing. A very short section (current section 3) describes the study area and data, and already includes some information on the methods. Section 4 is the detailed section on the methods. Section 5 shows the main results with a focus on the following points: the accuracy of the OBIA workflow proposed, including the influence of variables chosen -brightness, NDVI, NDWI, SD_NDWI- on accuracy, on the one hand, and the results on (i) topographic factors that influence avalanche activity, (ii) density of avalanches and (iii) classification in release, flow and run-out zone of detected avalanches on the other hand. Section 6 is a discussion on the advantages and drawbacks (points to be improved) regarding the method proposed. Finally, the last section concludes by mostly giving a summary of the main points discussed in section 6.

The paper presents an interesting method for detection of avalanche activity, which is based on object-based image analysis (OBIA) of near-infrared (NIR) aerial imagery. Given that NIR aerial images of good resolution are available, this method has the potential to cover large-scale mountainous areas including very remote areas. Furthermore, a statistical approach allows to distinguish between release, flow and run-out zones of the detected avalanches. One tricky yet important issue remains the distinction between single and multiple events. The OBIA workflow and the results presented in the paper are worth to be published in NHESS. However, the paper in its current form is not ready for publication. Some substantial revision is needed at least for two main reasons. First, I found that the sensitivity analysis of the method and outcomes to varying the different thresholds is lacking. The authors should pay attention to including such a sensitivity analysis when possible, and/or argue more on their choice regarding the thresholds of a number of parameters: see my specific comments below. Second, the authors should make an effort to improve the editing/structure of their manuscript that is sometimes quite hard to follow: short section versus another much longer section, announcement of outlines needed in the main introduction and the long section on "methods", etc. (see my comments regarding Editing/Structure of the paper and technical corrections).

*We would like to thank for this comment. We will take all the suggestions provided by the Reviewer to improve the quality of our article. We will focused in the revised manuscript on improving the description of sensitivity analysis which we used in our approach. We had long discussions within the author team about the best structure, however we will try one more time to make the structure easier to read. The further specific issues which we will take into consideration are described and answered in our further responds to the reviewer comments.*

Specific comments:

- section 1 (intro), page 2 (lines 20-21): Saying "..., focussing mainly on geographic coordinates, but rarely on any detailed information about their extent or area." Appears too strong to me. I think the authors should qualify that statement. Traditional methods, such as photo-interpretation, the use of testimonies, photographs, etc, gives crucial information, and merging that information into one single platform (see for instance the paper by Irstea group, by Bourova et al. CRST 2016, as well as some references therein) is already an important and efficient step. Could you please revise this part of the text?

*We agree with the Reviewer that this sentence might be misleading for the reader. We agree that photo-interpretation allows us to obtain detailed information about the shape and the size of avalanches. Our intention in this sentence was to stress that at the country level (e.g. in Switzerland) the information about snow avalanches are usually collected as XY coordinates of the occurred event, and this information can be obtained without any remote sensing data. However, such information are added to the snow avalanche database mostly only when the event produce a material damage or is a cause of an accident. As Reviewer noticed more detailed information about the size and the shape of the event can be collected for areas where a detailed remote sensing data are available. However, still collecting an aerial images in the winter season is not preferable (what we have mentioned in our article on Page 2, line 28) therefore obtaining such detailed information at the country level might be challenging.*

*We will restructure the sentence in our manuscript to avoid misunderstandings of its interpretation.*

- section 2, page 3 (lines 23-25): Please could you argue a little bit on this 35 deg threshold? I believe such a limit should depend on the snow type and then influence the results of your automatic detection approach. Please could you comment on this point?

*We would like to thank for this comment. However, we would like to point out that in our manuscript we did not applied 35 deg threshold. Additionally, Figure 7 in our article shows clearly that the avalanches which we have mapped manually and used as a reference data occur also in slope >35 deg. This threshold has been used in a publication which we are citing in these lines. In this article (Bühler et al. 2009) is also mentioned that such approach may misclassify some upper parts of avalanches, therefore in the OBIA approach which we are presenting herein we did not use any slope based assumption.*

- section 2, page 4 (line 25): again, this threshold of 35 deg needs much more discussion. This angle should depend on the snow type. Either you show a sensitivity analysis of your detection to that threshold, or you give more physical arguments on this choice.

*Same as above, we did not applied 35 deg assumption. This threshold has been used by Vickers et al. 2016, and in this sentence we are only explaining the method which has been used by these authors to extract snow avalanches.*

- section 4.1.1, page 6 (line 4): why this threshold of 6.25 square meters (the exponent 2 for the segment area is missing)? Could you please argue to make your choice less arbitrary? Did you conduct any sensitivity analysis of your method to varying that threshold? If yes, could you please include a thorough discussion on that analysis in the manuscript?

*The exponent 2 for segment is fine, we checked this in an online version of the article and it appears fine there. The 6.25 square meters threshold is selected by using trial-and-error approach which we have performed in the eCognition software by visual interpretation of the results when changing the threshold. Applying this threshold allowed us to obtain the highest correctness in classifying part of avalanches which brightness did not fulfilled the threshold assumption for snow avalanches, but which is fact were parts of snow avalanches.*

- section 4.1.2, page 6 (lines 13-16): the sentence is not clear to me, please revise. Could you please show a couple of examples of the cross-comparison between segment values and their visual representation in an image?

*Thank you for this notice, we agree that this sentence may not be clear. What we wanted to say is that inside some segments which have been classified as avalanches we have noticed small areas with smooth snow, which have been omitted by the avalanches, so they do not represent an avalanche and should be reclassified. We will rephrase this sentence to make it clearer. We will also add some example of cross-comparison between segment values and their visual representation in Figure 4.*

- end of section 4.1.2: could you please argue on this threshold of 62.2 square meters? Again: did you conduct any sensitivity analysis? Also I would suggest you show a square of that size on one figure (among figs a,b,c,d shown in Fig. 4 for instance). It would be useful for the reader to materialize the physical size of that threshold onto the map, compared to avalanche extensions that are detected.

*The threshold 62.2 square meters has been selected by performing the sensitivity analyses in the eCognition software, by performing the visual interpretation of the results when changing the thresholds. As suggested by the reviewer we will add a square in Figure 4 to show the size of this threshold with respect to the size of classified avalanches.*

- section 4.1.4, page 7 (line 4): could you please explain why you are using/choosing those values for both brightness and NDVI thresholds? I would suggest you to show a sensitivity analysis to varying those thresholds. This seems to me a crucial point if you would like your method to be possibly extended to much larger scales or other mountainous areas.

*We used these thresholds because they fitted well in separating our data into those representing snow and those that do not represent snow. The sensitivity analyses for Brightness and NDVI are already presented in Figure 6.*

- section 4.1.5, page 7 (lines 24-25 in brackets): why those thresholds? Could you show a sensitivity analysis to varying the thresholds? Brightness threshold is 2500 here, while it was 3000 a little earlier in the text (see previous comment too).

*As we mentioned above the sensitivity analyses are shown in Figure 6. The thresholds in this sentences (and this stage of classification) are different because some parts of snow avalanches were a bit darker or a little vegetated, so applying the previous threshold assumptions did not allow their correct assignment to an avalanche class. Therefore, we lowered these thresholds based on the sensitivity analyses presented in Figure 6 to allow their correct classification.*

- section 5, page 8 (lines 17-22): this part discusses the problems/limits of the method. The reader would like to know how those problems/limits are sensitive to the choice of the different thresholds (see the specific comments above). Could you please strengthen the discussion on this point?

*Thank you for this valuable comment. We will add more description on limits and problems that occur during the classification process.*

- section 5.3, page 9 (line 12): is the range 20-40 deg compatible with the threshold of 35 deg discussed in section 2?

*No, they are not compatible. Our results shows clearly that avalanches can occur also in slope >35 deg, therefore, in our OBIA approach we did not used any slope based assumptions to avoid rejection of some avalanches occurring on slope >35 deg what have been an issue in articles published by Bühler et al. 2009 and Vickers et al. 2016.*

Editing/Structure of the paper and Technical corrections (typings errors, etc.):

- Abstract (line 19): "... at three 4-km areas..." The exponent 2 for areas in square kilometres per square is missing?

*We have checked this issue in the whole manuscript submitted to the NHESS journal carefully and could not reproduce these errors.*

- end of section 1 (intro), page 3 (lines 14-15): again, the exponent 2 is missing here. Please fix it.

*Please see our response above.*

- section 1: would be nice to terminate the section by announcing the detailed structure of the paper, with a short summary of each section (including explicit numbered reference to each section).

*We did discuss the possibility of an additional short summary within the author team. But we agreed that it is not necessary and makes the paper just longer.*

- section 2, page 3 (lines 23-25): the construction of the sentence is a bit weird... The threshold of 35 deg is an assumption (a model input that stems from the DEM data) but not an outcome of the RAMMS model. Please revise the sentence.

*Thank you for this notice. We will restructure this sentence to avoid misunderstandings.*

- section 2, page 3 (line 27): "nadir?". Please fix this.

*Yes, the word "nadir" is fine.*

- section 3, page 5 (line 10): again, the exponent 2 is missing... PLEASE CHECK CAREFULLY the entire text regarding this issue, which is present in many parts of the manuscript.

*As we mentioned above, we think that this issue did not come up on our side. In the whole manuscript submitted to the NHESS journal all exponent 2 are fine.*

- section 3 is very very short. Given the fact that it already includes some information on the methods, I would suggest that the authors include section 3 as a sub-section of section 4 "Methods".

*We agree that this section is very short; however, we would be far away from merging it with methods section, which as reviewer mentioned is very long. Firstly, these two sections deal with other issues: 1) research area and data, 2) methods. Secondly, it would lengthens the 4 chapter which is already long and has many subchapters.*

- introduction of section 4: this part needs careful revision. Would be nice to put here an outline with explicit references to the sub-sections that follow (by using the numbering), as well as explicit reference to key figure 4. In its current form, I must say that section 4 is very difficult to read/follow.

*Thank you for this notice. We will add to the text the numbering of subchapters to make the text easier to read.*

- end of section 4.1.2, page 6 (line 19): typo... should be "then" instead of "than"

*Thank you for this remark. We will change "than" to "then".*

- section 5.4: I would suggest to replace "inset 1, Fig. 8" by "inset on Fig. 8a" and "inset 2 on Figure 8" by "inset on Fig 8b".

*Thank you for this suggestion. We will change it accordingly.*

O. Jaquet (Referee)
olivier.jaquet@in2earth.com

This paper presents an automatic object-based image analysis approach for the detection of snow avalanches using specific remote sensing data. The approach is explained in details and some statistical tests are performed for the evaluation of results uncertainty in relation to a reference data set. Following additional explanations/ comments would be valuable for the reader:

I.     A description of the conceptual model associated to the OBIA approach, it would provide some insight in relation to uncertainties and limitations; since only 61% of the total avalanche area was correctly identified...

*Thank you for this valuable comment. We will expand the limitations and the conceptual model of the OBIA approach in the discussion section of our manuscript. The 61% of the total avalanche area which have been correctly classified comes mostly from the fact that many of avalanches which have been classified in our area are not fresh, so their properties differ significantly from the properties of fresh avalanches what makes their simultaneous classification more difficult.*

II.     Mathematical details of the selected probability approach; to my understanding an uniform distribution is selected in relation to the elevation; the selection of such simple statistical model should be justified in relation to point I.

*Unfortunately, we do not understand the point in this comment. We would be grateful for providing more detail information in this matter.*

III.     A reference for confusion matrices

*The reference for confusion matrices which we used are provided in the article on page 8, lines 7-9: "We used the reference data to estimate several classification accuracy metrics, including Type I, Type II, and total errors (Sithole and Vosselman, 2004), overall, user's, and producer's accuracies (Congalton, 1991), Cohen's kappa (Cohen, 1960), and F-Score"*

IV.     The impact of snow thickness on the results

*In our article we did not analysed the impact of snow thickness on the results, because the accuracy of snow thickness map which we have in our disposal was not enough good to use it for analysis. However, we will add a sentence in this matter in the discussion section of our article.*

V.     The potential use of additional data (radar, seismic, ...) to improve the results

*We are not sure if we understand the comment correctly. In general, we think that the radar data and other seismic data give an opportunity in detecting snow avalanches, about what we have mentioned in the introduction chapter by citing the work published by Eckerstorfer and Malnes 2015. We think that, combining the results obtained from aerial images and radar data may increase the accuracy of hazards maps. However, we are a bit sceptic by combining aerial images and radar data in classifying snow avalanched with OBIA approach, especially because the collection of these data would have to be synchronized what may be difficult to do and expensive.*

---

## Author Response (AR1)

Department of 3D Mapping
BSF Swissphoto GmbH
Mittelstrasse 7
12529 Schönefeld, Germany

Schönefeld, 30[th] August 2017

Dear Editor,

We are pleased to submit our replies to the Reviewer's comments of manuscript NHESS-2017-120 entitled "Regional snow-avalanche detection using object-based image analysis of near-infrared aerial imagery" article.

We would like to thank two reviewers for their constructive comments, and appreciate their insightful analysis of our study. We considered all comments to improve the quality of the article. Following the suggestions, we rephrased potentially misleading statements, and added more descriptions of the sensitivity analyses and the thresholds that we have selected in our approach. We also elaborated more on the limits and problems tied to the classification protocol. We further expanded the limitations and the conceptual model of the OBIA approach in the discussion section. Finally, we corrected the figures according to a reviewer's request, and checked the text for spelling mistakes and clarity of language.

For the convenience, we address the reviewers' comments as outlined below:
- Reviewer's comments (**bold**),
- Our answers (*italic*), and the changes made in the text (*italic*).

Yours sincerely,

Karolina Korzeniowska
(on behalf of co-authors)

Anonymous Referee #1

General comments:

The paper by K. Korzeniowska and co-workers describes a method for automatic avalanche detection based on object-based image analysis (OBIA) of near-infrared (NIR) aerial imagery. The paper starts with an introduction of the dangers and risks related to snow avalanches with some detailed information on human fatalities and damages to infrastructures (mostly focused on the case of Switzerland), and the need for further developments concerning the large-scale yet precise mapping of avalanche activity. Section 2 provides the reader with an overview of the existing studies on automatic detection of avalanche activity based on image processing. A very short section (current section 3) describes the study area and data, and already includes some information on the methods. Section 4 is the detailed section on the methods. Section 5 shows the main results with a focus on the following points: the accuracy of the OBIA workflow proposed, including the influence of variables chosen -brightness, NDVI, NDWI, SD_NDWI- on accuracy, on the one hand, and the results on (i) topographic factors that influence avalanche activity, (ii) density of avalanches and (iii) classification in release, flow and run-out zone of detected avalanches on the other hand. Section 6 is a discussion on the advantages and drawbacks (points to be improved) regarding the method proposed. Finally, the last section concludes by mostly giving a summary of the main points discussed in section 6.

The paper presents an interesting method for detection of avalanche activity, which is based on object-based image analysis (OBIA) of near-infrared (NIR) aerial imagery. Given that NIR aerial images of good resolution are available, this method has the potential to cover large-scale mountainous areas including very remote areas. Furthermore, a statistical approach allows to distinguish between release, flow and run-out zones of the detected avalanches. One tricky yet important issue remains the distinction between single and multiple events. The OBIA workflow and the results presented in the paper are worth to be published in NHESS. However, the paper in its current form is not ready for publication. Some substantial revision is needed at least for two main reasons. First, I found that the sensitivity analysis of the method and outcomes to varying the different thresholds is lacking. The authors should pay attention to including such a sensitivity analysis when possible, and/or argue more on their choice regarding the thresholds of a number of parameters: see my specific comments below. Second, the authors should make an effort to improve the editing/structure of their manuscript that is sometimes quite hard to follow: short section versus another much longer section, announcement of outlines needed in the main introduction and the long section on "methods", etc. (see my comments regarding Editing/Structure of the paper and technical corrections).

*We thank the Reviewer for this positive assessment. We took into account all the suggestions provided by the Reviewer to improve the quality of our manuscript. Specifically, we focused in our revisions on improving the description of sensitivity analysis. We had long discussions within the author team about the best structure, and aimed for a more accessible structure of the workflow. Please see below for our specific responses:*

Specific comments:

- section 1 (intro), page 2 (lines 20-21): Saying "..., focussing mainly on geographic coordinates, but rarely on any detailed information about their extent or area." Appears too strong to me. I think the authors should qualify that statement. Traditional methods, such as photo-interpretation, the use of testimonies, photographs, etc, gives crucial information, and merging that information into one single platform (see for instance the paper by Irstea group, by Bourova et al. CRST 2016, as well as some references therein) is already an important and efficient step. Could you please revise this part of the text?

*We agree with the Reviewer that this sentence might be misleading. We agree that photo-interpretation allows us to obtain detailed information about the shape and the size of avalanches. Our intention here was to stress that at the national level (e.g. in Switzerland) information about snow avalanches is usually collected as geographic point coordinates; this information can be obtained without any remote sensing data. However, coordinates are added to snow-avalanche databases mostly only in case of damage or accident. The Reviewer noted that more detailed information about avalanche geometry can be collected for areas where a detailed remote sensing data are available. However, using aerial images in the winter season is problematic, as we pointed out on page 2, lines 30-33), so that collecting detailed information at the national level remains challenging.*

*We rephrased the sentence in our manuscript to avoid misunderstandings of its interpretation as follow, page 2, lines 20-21:*

*"To date, experts map most avalanches manually, and at the national level mainly the geographic coordinates of avalanches; rarely does this mapping involve detailed information about the avalanche geometry (Bühler et al., 2009)."*

- section 2, page 3 (lines 23-25): Please could you argue a little bit on this 35 deg threshold? I believe such a limit should depend on the snow type and then influence the results of your automatic detection approach. Please could you comment on this point?

*We point out that this slope threshold is not part of our study, so that we do not elaborate on that. Figure 7 shows that the avalanches that we mapped manually and used as reference data occur also on steeper slopes. Bühler et al. (2009) noted that using a slope threshold may misclassify some detachment areas of avalanches; therefore we excluded any assumptions regarding slope in our OBIA approach.*

- section 2, page 4 (line 25): again, this threshold of 35 deg needs much more discussion. This angle should depend on the snow type. Either you show a sensitivity analysis of your detection to that threshold, or you give more physical arguments on this choice.

*Please see our reply above: we did not use any slope threshold in our study. We were referring to its use in previous studies.*

- section 4.1.1, page 6 (line 4): why this threshold of 6.25 square meters (the exponent 2 for the segment area is missing)? Could you please argue to make your choice less arbitrary? Did you conduct any sensitivity analysis of your method to varying that threshold? If yes, could you please include a thorough discussion on that analysis in the manuscript?

*The exponent 2 for segment is fine, we checked this in an online version of the article and it appears fine there. We selected the 6.25 square meters threshold by a trial-and-error approach in the eCognition software by visual checking the results for various thresholds. Applying this threshold allowed us to obtain the highest correctness in classifying those parts of avalanches that did not meet the brightness threshold of snow avalanches. We made this clearer on page 6, lines 12-14:*

*"To select the best size of segments we ran a sensitivity analysis using a trial-and-error approach in the eCognition software, visually checking the results for various thresholds."*

- section 4.1.2, page 6 (lines 13-16): the sentence is not clear to me, please revise. Could you please show a couple of examples of the cross-comparison between segment values and their visual representation in an image?

*We agree that this sentence may not be fully clear. What we wanted to say is that inside some segments that we classified as avalanches we noticed small areas of smooth snow not representing avalanche debris and thus in need of reclassification. We rephrased this sentence to make it clearer, and added some example of cross-comparison between segment values and their visual representation in Figure 4. See page 6, lines 12-16:*

*"We thus included many pixels as parts of avalanche deposits that in the previous step had escaped being classified as 'rough snow' because of $SD_{NDWI}$ values. We set a maximum area of segment to be reclassified as 'rough snow' to avoid including large, but smooth, areas inside avalanches that are not avalanche debris."*

- end of section 4.1.2: could you please argue on this threshold of 62.2 square meters? Again: did you conduct any sensitivity analysis? Also I would suggest you show a square of that size on one figure (among figs a,b,c,d shown in Fig. 4 for instance). It would be useful for the reader to materialize the physical size of that threshold onto the map, compared to avalanche extensions that are detected.

*We selected the threshold of 62.2 square meters after running a sensitivity analyses in the eCognition software, and visually checking the results when changing the thresholds. Following the Reviewer's suggestion, we added squares in Figure 4 to show the size of this threshold with respect to the size of classified avalanches. Please see the changes in the text on page 6, lines 29-31:*

*"We chose this area threshold based on visual checks and a trial-and-error sensitivity analysis in the eCognition software."*

- section 4.1.4, page 7 (line 4): could you please explain why you are using/choosing those values for both brightness and NDVI thresholds? I would suggest you to show a sensitivity analysis to varying those thresholds. This seems to me a crucial point if you would like your method to be possibly extended to much larger scales or other mountainous areas.

*We used these thresholds because they fitted well in separating our data into those representing snow and those that do not represent snow. The sensitivity analyses for Brightness and NDVI are already presented in Figure 6. We added additional sentence in this matter, please see page 7, lines 12-14:*

*"The brightness thresholds we derived from a sensitivity analysis, in analogy to previous steps (Fig. 6)."*

- section 4.1.5, page 7 (lines 24-25 in brackets): why those thresholds? Could you show a sensitivity analysis to varying the thresholds? Brightness threshold is 2500 here, while it was 3000 a little earlier in the text (see previous comment too).

*Please see our reply above; the results of the sensitivity analyses are shown in Figure 6. The thresholds in this sentence (and this stage of classification) are different because some parts of snow avalanches were a bit darker or a little vegetated, so applying the previous threshold assumptions did not allow their correct assignment to an avalanche class. Therefore, we lowered these thresholds based on the sensitivity analyses in Figure 6 to allow for correct classification. We added a sentence on page 8, lines 4-6:*

*"The threshold selected for brightness is lower than in previous steps, because some parts of snow avalanches were a little darker; hence using the same threshold would cause unwanted misclassifications."*

- section 5, page 8 (lines 17-22): this part discusses the problems/limits of the method. The reader would like to know how those problems/limits are sensitive to the choice of the different thresholds (see the specific comments above). Could you please strengthen the discussion on this point?

*We added more detail about the limits and problems that occur during the classification process to the discussion section, please see the changes on page 12, lines 27-35:*

*"The most difficult to classify were old avalanches where the bare ground cropped out or where vegetation occurred in the path of the avalanche. These avalanches did not meet the assumptions in our OBIA protocol and could not be classified correctly because they were not rough enough, or were too vegetated or dark due to thin snow cover. Here a solution may be using different thresholds for old and fresh avalanches to optimise our OBIA approach. We tested different thresholds for the input layers and different neighbourhood assumptions in order to include these avalanches; however, this resulted in more false positives, so the cost of correct classification of these avalanches was higher than the benefit. We therefore decided to stay with the same workflow and thresholding shown in Figure 4 for the whole study area."*

- section 5.3, page 9 (line 12): is the range 20-40 deg compatible with the threshold of 35 deg discussed in section 2?

*No, they are not compatible. Our results show that avalanches can occur also in slopes steeper than 35 degree. Coming back to our reply to a similar comment, we note that we did not incorporate any specific slope thresholds. We added a sentence in this matter, please see page 9, lines 14-17:*

*"Our results show that avalanches were also detected on slopes steeper than 35°, thus highlighting the limits of detection methods using arbitrary slope thresholds (Bühler et al., 2009; Vickers et al., 2016)."*

Editing/Structure of the paper and Technical corrections (typings errors, etc.):
- Abstract (line 19): "... at three 4-km areas..." The exponent 2 for areas in square kilometres per square is missing?

*We have checked this issue in the whole manuscript carefully and could not reproduce these errors.*

- end of section 1 (intro), page 3 (lines 14-15): again, the exponent 2 is missing here. Please fix it.

*Please see our response above.*

- section 1: would be nice to terminate the section by announcing the detailed structure of the paper, with a short summary of each section (including explicit numbered reference to each section).

*We did discuss the possibility of an additional short summary within the author team, but agreed that it is not necessary and would make the paper unduly longer.*

- section 2, page 3 (lines 23-25): the construction of the sentence is a bit weird... The threshold of 35 deg is an assumption (a model input that stems from the DEM data) but not an outcome of the RAMMS model. Please revise the sentence.

*We restructured this sentence to avoid misunderstandings. Please see page 3, lines 28-30:*

*"They used the numerical simulation tool RAMMS (Rapid Mass Movement Simulation; Christen et al., 2010) to identify possible avalanches, and excluded slopes >35° from the runout calculation, assuming that these slopes could not accumulate snow-avalanche debris."*

- section 2, page 3 (line 27): "nadir?". Please fix this.

*Yes, the word "nadir" is fine.*

- section 3, page 5 (line 10): again, the exponent 2 is missing... PLEASE CHECK CAREFULLY the entire text regarding this issue, which is present in many parts of the manuscript.

*As we mentioned above, we think that this issue did not come up on our side. In the entire manuscript submitted all exponents are fine.*

- section 3 is very very short. Given the fact that it already includes some information on the methods, I would suggest that the authors include section 3 as a sub-section of section 4 "Methods".

*We agree that this section is very short, but would maintain its structure focusing on some basics of the study area.*

- introduction of section 4: this part needs careful revision. Would be nice to put here an outline with explicit references to the sub-sections that follow (by using the numbering), as well as explicit reference to key figure 4. In its current form, I must say that section 4 is very difficult to read/follow.

*We added to the text the numbering of subchapters to make the text easier to read, and also the numbering of subchapters to Figure 4. Please see the changes in the text on page 5, lines 14-20:*

*"We introduce an automatic method for mapping release zones, tracks, and runout zones of avalanches using NIR 0.25-m aerial images (Section 4.1). We compare the automatic classification with manually digitised reference data and estimated the accuracy of detecting snow avalanches with confusion matrices (Sections 4.2, 5.1). We also investigated the topographical conditions in which most mapped avalanches occurred (Section 5.3), using two approaches for visualising avalanche density (Section 5.4). We also propose a probability approach to representing release and runout zones of avalanches and the automatic classification of snow avalanche parts (Section 5.5.)."*

- end of section 4.1.2, page 6 (line 19): typo... should be "then" instead of "than"

*OK, we changed "than" to "then".*

- section 5.4: I would suggest to replace "inset 1, Fig. 8" by "inset on Fig. 8a" and "inset 2 on Figure 8" by "inset on Fig 8b".

*We changed this accordingly.*

O. Jaquet (Referee)

olivier.jaquet@in2earth.com

This paper presents an automatic object-based image analysis approach for the detection of snow avalanches using specific remote sensing data. The approach is explained in details and some statistical tests are performed for the evaluation of results uncertainty in relation to a reference data set. Following additional explanations/ comments would be valuable for the reader:

I.      A description of the conceptual model associated to the OBIA approach, it would provide some insight in relation to uncertainties and limitations; since only 61% of the total avalanche area was correctly identified…

*We elaborated on the limitations and the conceptual model of the OBIA approach in the discussion section of our manuscript. The 61% of the total avalanche area correctly classified comes mostly from the observations that many avalanches classified were not fresh, so their properties differed from those of fresh avalanches. We addressed this matter on page 12, lines 22-27:*

*"Our method correctly identified 61% of the total avalanche area in the study area, mainly because of heterogeneous avalanche debris with dark or smooth patches, and because of differing deposit ages, and hence differing surface roughness."*

II.     Mathematical details of the selected probability approach; to my understanding an uniform distribution is selected in relation to the elevation; the selection of such simple statistical model should be justified in relation to point I.

*Unfortunately, we do not fully understand the point of this comment. We would be grateful for providing more detail information in this matter.*

III.    A reference for confusion matrices

*The reference for confusion matrices which we used are provided in the manuscript on page 8, lines 19-22:*

*"We used them to estimate several performance metrics, including Type I, Type II, and total errors (Sithole and Vosselman, 2004), overall, user's, and producer's accuracies (Congalton, 1991), Cohen's kappa (Cohen, 1960), and F-Score"*

IV.     The impact of snow thickness on the results

*We did not analyse the impact of snow thickness on the results, mainly limited by the low accuracy of available snow-thickness maps. However, we added a sentence to the discussion section, please see page 13, lines 11-14:*

*"Detailed enough maps of snow thickness may help to improve the classification accuracy, and perhaps enable separating release zones, tracks, and runout zones from the surrounding snow cover."*

V.      The potential use of additional data (radar, seismic, …) to improve the results

*We are not sure if we understand this comment correctly. In general, we think that the radar data and other seismic data provide further opportunities for detecting snow avalanches, and we mentioned*

*that in the introduction chapter by citing Eckerstorfer and Malnes (2015). We think that, combining the results obtained from aerial images and radar data may increase the accuracy of hazards maps. However, we are a bit sceptic by combining aerial images and radar data in classifying snow avalanches with OBIA, especially because the collection of these data would have to be synchronized. We added a sentence to the discussion section, please see page 13, lines 22-23:*

[revised manuscript text omitted]